# SelfPrompt: Confidence-Aware Semi-Supervised Tuning for Improved Vision-Language Model Adaptation

**Shuvendu Roy**                                                    *shuvendu.roy@queensu.ca*
*Queen's University, Canada*

**Ali Etemad**                                                          *ali.etemad@queensu.ca*
*Queen's University, Canada*

**Reviewed on OpenReview:** *https://openreview.net/forum?id=cP6USDUjK8*

## Abstract

We present SelfPrompt, a novel prompt-tuning approach for vision-language models (VLMs) in a semi-supervised learning setup. Existing methods for tuning VLMs in semi-supervised setups struggle with the negative impact of the miscalibrated VLMs on pseudo-labelling, and the accumulation of noisy pseudo-labels. SelfPrompt addresses these challenges by introducing a cluster-guided pseudo-labelling method that improves pseudo-label accuracy, and a confidence-aware semi-supervised learning module that maximizes the utilization of unlabelled data by combining supervised learning and weakly-supervised learning. Additionally, we investigate our method in an active semi-supervised learning setup, where the labelled set is strategically selected to ensure the best utilization of a limited labelling budget. To this end, we propose a weakly-supervised sampling technique that selects a diverse and representative labelled set, which can be seamlessly integrated into existing methods to enhance their performance. We conduct extensive evaluations across 13 datasets, significantly surpassing state-of-the-art performances with average improvements of 6.23% in standard semi-supervised learning, 6.25% in active semi-supervised learning, and 4.9% in base-to-novel generalization, using a 2-shot setup. Furthermore, SelfPrompt shows excellent generalization in single-shot settings, achieving an average improvement of 11.78%.

## 1 Introduction

Vision-language models (VLMs) (Radford et al., 2021) pre-trained on large-scale datasets of image-text pairs have shown strong generalization on a wide range of tasks. Nonetheless, prior works (Zhou et al., 2022a;b) have demonstrated that VLMs require fine-tuning on a considerable amount of labelled data to perform well on downstream tasks. Additionally, the size of the foundation model makes fine-tuning in a limited labelled data setting difficult without losing generalization (Roy & Etemad, 2024). To reduce the reliance on labelled data, some recent works have explored solutions that utilize auxiliary unlabelled data (Menghini et al., 2023; Zhang et al., 2024) to improve learning from a limited set of labelled data.

Although prior works that leverage unlabelled data for tuning VLMs show substantial performance gains, we identify several limitations in such approaches. **(a)** Given the unlabelled set, prior works (Menghini et al., 2023; Zhang et al., 2024) utilize the zero-shot capabilities of pre-trained VLMs to predict pseudo-labels for the unlabelled data to then use as labelled samples. However, pre-trained VLMs do not necessarily possess adequate knowledge of the downstream domain, which leads to incorrect pseudo-labels. Such wrong labels, especially in few-labelled settings, can negatively impact the final performance of the model. **(b)** To learn from unlabelled data more effectively, previous works (Menghini et al., 2023; Zhang et al., 2024) have employed incremental pseudo-labelling, wherein the labelled set is continuously expanded by iteratively adding to the pseudo-label set from the unlabelled set. Nevertheless, as Figure 1 illustrates, this method often results in the accumulation of noisy pseudo-labels, ultimately leading to performance degradation.

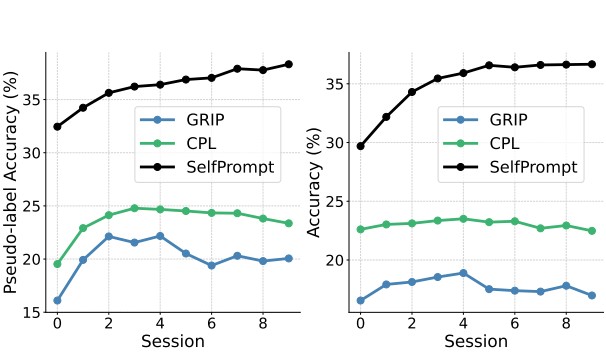 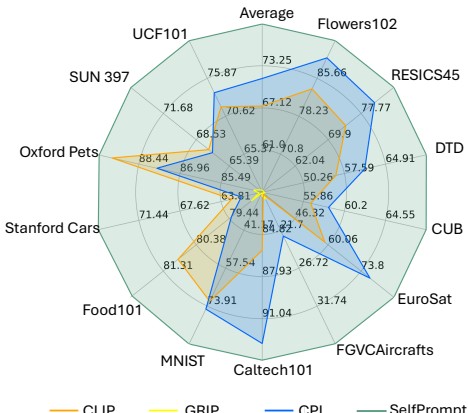

Figure 1: (left) Pseudo-label accuracy; (right) Test accuracy over training sessions.

Figure 2: Performance comparison to prior works on semi-supervised tuning of VLMs.

To solve the above-mentioned problems, we propose **SelfPrompt**, a new prompt tuning approach that leverages weak supervision by the pre-trained VLM to fine-tune itself with confidence-aware semi-supervised learning. Instead of using the VLM's predictions as pseudo-labels, which are often noisy, we introduce a novel cluster-guided strategy for pseudo-labelling. Specifically, our method clusters all labelled and unlabelled samples in the embedding space, using the labelled set as the cluster centres. Unlabelled samples near the centroids of these clusters are then selected and assigned the corresponding class labels to form the pseudo-labelled set. Furthermore, to prevent noise accumulation from relying solely on pseudo-labels during training, we adopt a hybrid approach that uses high-confidence pseudo-labels in a fully supervised setting, while learning from low-confidence samples in a weakly supervised manner.

In a standard semi-supervised learning setup, we are provided with a limited labelling budget (few samples per class), where existing methods (Zhang et al., 2024; Menghini et al., 2023) typically select a random subset from the unlabelled set for labelling. However, such random selection may fail to adequately represent the underlying data distribution, leading to an inefficient labelling strategy. In this work, we explore **active semi-supervised learning**, where a subset is strategically selected from the unlabelled set to form the labelled set. To achieve this, we propose weakly-supervised sampling, a novel approach that can be integrated with any existing semi-supervised learning method for selecting a diverse and representative subset of samples from the unlabelled data. Our weakly-supervised sampling follows a two-step protocol. First, predictions from a VLM (vision-language model) are used as a source of weak supervision to filter out both the most and least confident samples from the unlabelled set. This is based on the observation that highly confident samples have low information gain and contribute little to learning, while the least confident samples are often noisy and unrepresentative of the dataset. Next, a clustering-based selection technique identifies a diverse set of samples from the remaining unlabelled data for labelling.

We present extensive evaluations of our proposed solution on both standard semi-supervised learning and active semi-supervised learning setups. While previous works (Menghini et al., 2023; Zhang et al., 2024) report these results on six datasets, we evaluate our solution (and the previous methods) on **13 datasets**. Our evaluation shows that SelfPrompt outperforms prior works by 6.23% on average in the standard semi-supervised learning setup and by 6.25% on the active semi-supervised learning setup, with up to 11.23% and 13.82% improvements on individual datasets, respectively. Additionally, our weakly-supervised sampling module shows considerable improvement when added to the existing methods. We further demonstrate the generalizability of our solution by reducing the size of the labelled set to just one sample per class while outperforming prior methods by an average of 11.78%. For completeness, we also compare our solution to few-shot tuning methods (Zhou et al., 2022a; Khattak et al., 2022; Roy & Etemad, 2024; Li et al., 2024) on base-to-novel generalization, where we also show an average improvement of 4.9% over prior works in a 2-shot setup, with improvements in both base and novel classes. Finally, we present extensive ablation and sensitivity studies on different components of our proposed method. Overall, our contributions are:

- We introduce SelfPrompt, a novel prompt-tuning approach for VLMs that uses weak supervision from the pre-trained VLM itself to learn effective representations from limited labelled data while improving generalization.

- We introduce three novel components: (a) Cluster-guided pseudo-labelling, which improves pseudo-label accuracy by assigning labels based on proximity to labelled samples; (b) Confidence-aware semi-supervised learning, which adapts learning based on the confidence of pseudo-labels to maximize the use of the unlabelled data; and (c) Weakly-supervised labelled set sampling that selects a representative and diverse set of labelled samples;

- Our comprehensive experiments across 13 datasets demonstrate that SelfPrompt surpasses existing methods on various benchmarks, achieving average improvements of 6.23% in standard semi-supervised learning, 6.25% in active semi-supervised learning, and 4.9% in base-to-novel generalization, in the standard 2-shot evaluation setup. Additionally, SelfPrompt exhibits strong generalization in a single-shot setting, resulting in an even higher improvement of 11.78%.

## 2 Related Works

**Vision-language models.** Vision-language models (VLMs), pre-trained on vast web-scale datasets of image-text pairs, have demonstrated remarkable generalization across a wide range of downstream tasks (Radford et al., 2021; Jia et al., 2021; Alayrac et al., 2022). These models are composed of a vision encoder and a text encoder, jointly trained to align the representations of an image and corresponding text. Once pre-trained, VLMs can perform zero-shot image classification by matching an image embedding with the text embedding of the class name with a prompt, such as 'a photo of a [class].' While this zero-shot capability has delivered impressive results in various domains, fine-tuning is often necessary to achieve optimal performance on new tasks or domains (Zhou et al., 2022a). However, fine-tuning large models with *limited* labelled data presents significant challenges, including overfitting and reduced generalization (Roy & Etemad, 2024). To address these issues, recent approaches have proposed semi-supervised tuning, which incorporates unlabelled data alongside limited labelled data to improve the fine-tuning of VLMs on downstream tasks. In the following subsections, we first explore prompt tuning, a prominent approach for parameter-efficient fine-tuning of pre-trained models by introducing a small set of additional parameters while keeping the pre-trained encoder frozen. These methods help mitigate the risk of overfitting on limited downstream labelled data. We then review prior work on the semi-supervised tuning of VLMs.

**Prompt tuning.** Prompt tuning is a parameter-efficient technique for adapting foundation models to downstream tasks by learning soft prompts (textual (Ge et al., 2023; Zhou et al., 2022c) or visual (Bahng et al., 2022a; Jia et al., 2022)) from limited labelled data (Ge et al., 2023; Jia et al., 2022). Text-based prompt tuning (Zhou et al., 2022c) optimizes learnable prompt vectors, which are embedded within the input sentence tokens and fed into the sentence encoder. Extending this idea, Zhou et al. (2022b) introduced soft prompt learning that conditions the prompt on the image input. In contrast, visual prompt tuning (Bahng et al., 2022b) introduces a small number of trainable parameters into the visual input tokens. PLOT (Chen et al.) fine-tuned VLM by learning multiple diverse prompts per category via a two-stage optimization strategy. MaPLe (Khattak et al., 2022) later introduced a multi-modal approach that trains both textual and visual prompts simultaneously, leveraging their synergy to facilitate multi-modal representation learning while avoiding an overemphasis on unimodal features. A similar multi-modal approach was proposed in PromptSRC (Khattak et al., 2023), which, unlike MaPLe, focused on learning task-agnostic and independent prompts for text and images. Later, CoPrompt (Roy & Etemad, 2024) proposed to enforce consistency between the pre-trained and learnable encoders, aiming to reduce overfitting and enhance generalization. AWT (Zhu et al., 2024) enhances vision-language models by augmenting inputs with diverse perspectives and enriched descriptions, dynamically weighting inputs by prediction uncertainty, and transporting semantic correlations into a shared embedding space. While these methods have demonstrated improved performance on downstream tasks compared to pre-trained VLMs, they still face challenges in domains where the target data significantly deviates from the pre-training distribution, primarily due to the limited availability of domain-specific data.

**Semi-supervised tuning.** Recently, a new stream of research has focused on semi-supervised tuning that leverages unlabelled data alongside a small set of labelled data to enhance downstream task performance. The core idea behind these methods is pseudo-labelling (Sohn et al., 2020; Lee et al., 2013), where the model predicts labels for unlabelled samples and then uses these pseudo-labels to learn from the unlabelled data.

For example, GRIP (Menghini et al., 2023) utilizes CLIP's zero-shot capabilities to generate pseudo-labels for unlabelled data and select the most confident samples to serve as labelled data. However, this approach introduces a considerable amount of wrong pseudo-labels due to the inherent miscalibration (LeVine et al., 2023) and imbalanced predictions (Wang et al., 2022a) issues of the pre-trained VLM. To address these issues, CPL (Zhang et al., 2024) proposes to generate refined candidate pseudo-labels through intra- and inter-instance label selection, using a confidence score matrix to improve label accuracy and class balance during fine-tuning. Both GRIP and CPL adopt an iterative process, where the model is used to continuously refine and select additional samples from the unlabelled set. A related approach, XPL (Chakraborty et al., 2024), introduces a cross-model framework in which a primary and an auxiliary network share the same frozen VLM but learn prompts of different lengths, using each other's confident predictions as pseudo-labels under a consistency regularization scheme. While XPL mitigates some noise through cross-model agreement, it still relies on the VLM's own predictions filtered by a fixed confidence threshold, inheriting the miscalibration problem, and does not consider the quality of the labelled set selection. Although these methods have demonstrated promising improvements in VLM performance on downstream tasks, they still face several key limitations. These include the under-utilization of the labelling budget, the negative impact of miscalibrated pseudo-labels, and the declining quality of pseudo-labelling as the number of samples increases.

**Weakly-supervised learning**. Weakly supervised learning encompasses a class of methods that train models with limited or imprecise supervision (Peyre et al., 2017; Li et al., 2019; Zhou, 2018). Unlike fully supervised learning, which requires large amounts of precisely labelled data, weakly supervised learning leverages weak annotations, such as noisy, incomplete, or coarse-grained labels. This paradigm significantly reduces the reliance on costly and time-intensive data annotation processes. This paradigm has demonstrated broad applicability across various domains, including vision-language models (Wang et al., 2022b), medical image analysis (Kanavati et al., 2020). Despite its potential, it has been underexplored in the context of semi-supervised learning, where pseudo-label predictions frequently introduce noise. This positions weakly supervised learning as an ideal candidate to address the challenges posed by noisy labels in semi-supervised frameworks.

## 3 Method

### 3.1 Preliminaries and Problem formulation

Let $\theta$ be a pre-trained image encoder and $\phi$ be a text encoder of a pre-trained VLM. For a given input image $x$, the VLM predicts the probability distribution over $C$ classes as:

$$p(y|x; \theta, \phi) = \frac{\exp(\text{sim}(z, w_y)/t)}{\sum_{k=1}^{C} \exp(\text{sim}(z, w_k)/t)}, \tag{1}$$

where $z = \theta(x)$ is the image embedding and $w_k$ is the class-embedding of class $k$, generated using a prompt template as $w_k = \phi(\text{'a photo of a } [\text{category}]_k\text{'})$, and $t$ is the temperature parameter. Since the zero-shot performance of the VLM is limited by its pre-trained knowledge of the downstream task, further fine-tuning is often necessary for the VLM to adapt effectively to new tasks or domains. A widely adopted approach for fine-tuning large foundation models is prompt tuning, which involves adding a small set of learnable parameters to the model while keeping the pre-trained encoder frozen. This approach typically involves prepending a set of $K$ learnable tokens, $P = \{p_1, p_2, \ldots, p_K\}$, to the tokenized input embeddings, enabling the model to adapt to new tasks with minimal parameter updates. Two common forms of prompt tuning are visual prompt tuning (Bahng et al., 2022b) and textual prompt tuning (Zhou et al., 2022a). We denote the prediction of VLM with the learnable tokens as $f(x) = p(y|x; \theta, \phi, P)$. A recent class of solutions proposes the use of semi-supervised learning for tuning VLMs by leveraging a large unlabelled set along with a small labelled set. In this setup, we are given a large unlabelled set $U = \{x_i\}_{i=1}^{M}$ with a label budget of $N \ll M$ samples. Here, $N = C \times n$, where $C$ is the number of classes, and $n$ is the samples per class.

Despite recent progress in semi-supervised prompt tuning for VLMs, we have identified two key open challenges in this area: **(a)** Given an unlabelled set $U$, the typical approach is to initiate the training protocol by using the pre-trained VLM to generate pseudo-labels for $U$ (using Eq. 1) and selecting the most confident $M/S$ samples (Zhang et al., 2024; Menghini et al., 2023). Here, $S$ represents the number of sessions

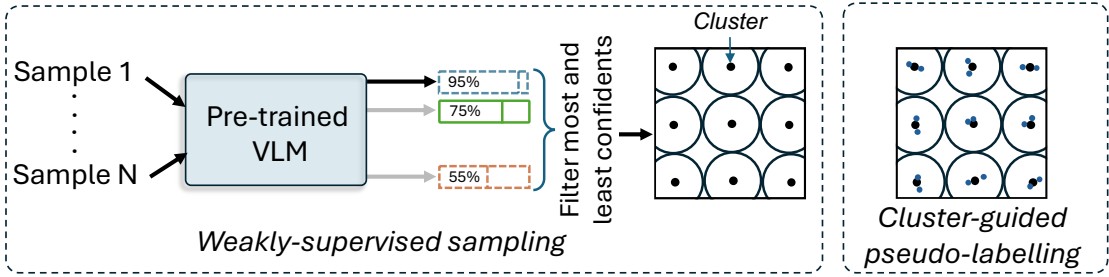

Figure 3: (left) A visual illustration of the weakly-supervised sampling module. Using predictions from the pre-trained VLM, the least and most confident samples, which are not representative of the downstream data, are filtered out. The remaining feature space is then clustered into a number of clusters equal to the labelling budget to ensure maximum diversity among the selected samples. (right) Cluster-guided pseudo-labelling assigns the same class label to samples near the cluster centers as the pseudo-label.

over which this pseudo-labelling and selection process is repeated. However, pre-trained VLMs often lack adequate knowledge of the target domain, i.e., domain miscalibration (LeVine et al., 2023), thus resulting in considerable numbers of incorrect pseudo-labels. This in turn can misguide the fine-tuning of the model, leading to degraded final performances. **(b)** To maximize the information obtained from unlabelled data, prior works have employed an iterative pseudo-labelling strategy, where unlabelled samples are progressively added to the pseudo-labelled set. However, as shown in Figure 1, this approach results in an accumulation of noisy samples, leading to a drop in the model's final performance. While CPL (Zhang et al., 2024) proposed avoiding pseudo-labels in favour of using multiple top predictions as partial labels, this results in under-utilizing the unlabelled data by avoiding correct pseudo-labels as well. Additionally, in a semi-supervised learning setting with $N$ samples, the common strategy is to randomly selected $n$ samples per class from the unlabelled set $U$ to form the labelled set (Zhang et al., 2024; Menghini et al., 2023). However, random selection may fail to adequately represent the data distribution of the target domain, leading to inefficient use of the limited label budget.

### 3.2 SelfPrompt

In light of the challenges above, we propose SelfPrompt, a novel semi-supervised prompt tuning method for VLMs. To improve the pseudo-label quality, especially at the beginning of the training, SelfPrompt introduces a novel cluster-guided pseudo-labelling approach that does not utilize the zero-shot prediction from the VLM as the pseudo-label. By not relying on VLM predictions for pseudo-labelling, we ensure that our method reduces reliance on direct zero-shot predictions as a source of pseudo-labels. Instead, we cluster all labelled and unlabelled samples in the embedding space, using the labelled set as the cluster centres: $\mathcal{C} = \{\mathcal{C}_1, \mathcal{C}_2, \ldots, \mathcal{C}_N\}$. In active semi-supervised setting, these clusters are formed using our weakly-supervised sampling module to be the most diverse and representative set of samples (Section 3.3). Since the clusters are formed based on embedding similarity, samples under the same cluster have similar semantics. Especially the samples close to the cluster centres (also close to the selected labelled sample) are likely to belong to the same class as the sample at the cluster's center. Implicating this realization, we select additional $p$ samples from each cluster and label them with the label of the cluster center. Specifically, for each cluster $\mathcal{C}_j$, we pick the $p$ samples closest the cluster centers to form a pseudo-label set $\mathcal{P}_j = \{x_j^1, x_j^2, \ldots, x_j^p\}$, where $\mathcal{P}_j$ is the pseudo-label set for cluster $\mathcal{C}_j$, and $x_j^k$ is the $k$-th closest sample to $x_j^*$. Here, closest sample is identified by $x_j^k = \arg\min_{x_i \in \mathcal{C}_j \setminus \{x_j^*\}} \|\mathbf{z}_i - \mathbf{z}_j^*\|^2$, where $\mathbf{z}_j^* = \theta(\mathbf{x}_j^*)$, and $\|\cdot\|^2$ is the squared Euclidean distance in the embedding space. Finally, each sample in $\mathcal{P}_j$ is assigned to the label of the cluster center of $\mathcal{C}_j$ to form our pseudo-label set $\mathcal{X}_p = \{(x_{j1}, y_j), (x_{j2}, y_j), \ldots, (x_{Np}, y_N)\}$.

Furthermore, to make the best use of the unlabelled data, we propose a confidence-aware semi-supervised solution that learns from the high-confident samples in a supervised learning setup, while learning from the low-confident samples in a weakly-supervised setting. Such a hybrid approach ensures efficient utilization of high-confident pseudo-labels while minimizing the adverse effects of low-confident samples. Specifically, we

first predict the output probability distribution for each sample in the unlabelled set $U$ as $\boldsymbol{p}_i = f(x_i) \in \mathbb{R}^C$. Then we incorporate the $t$ (defined as $\tau \times M$) most confident samples-per-class into our pseudo-label set:

$$\mathcal{X}^+ = \mathcal{X}_P \cup \big( \bigcup_{c=1}^{C} \text{top}_t(\{\boldsymbol{x}_i | \arg\max(\boldsymbol{p}_i) = c\}) \big), \tag{2}$$

where $\tau$ is a hyper-parameter that controls the number of samples to be included in the pseudo-label set, $\text{top}_t(\{x_i | \arg\max(\boldsymbol{p}_i) = c\})$ selects the $t$ most confident samples for each class $c$, with confidence defined as $max(\boldsymbol{p}_i)$. We learn from the remaining relatively low-confidence samples in a weakly-supervised setting. Specifically, we follow CPL (Zhang et al., 2024), and gather the top-k predictions per sample to form a weakly-labelled set $\mathcal{X}_{\text{weak}} = \{(x_i, s_i) \mid x_i \in U \setminus \mathcal{X}^+\}$, where $\boldsymbol{s}_i$ is a one-hot vector containing $\boldsymbol{s}_i^c = 1$ if class $c$ is among the top predictions for sample $i$. Finally, we learn from the labelled set $\mathcal{X}_L$, pseudo-labelled set $\mathcal{X}^+$, and weakly labelled set $\mathcal{X}_{weak}$, together as follow:

$$\mathcal{L}_{final} = \frac{1}{|\mathcal{X}_L|} \sum_{(x,y) \in \mathcal{X}_L} \ell(f(x), y) + \frac{1}{|\mathcal{X}_+|} \sum_{(x,y) \in \mathcal{X}_+} \ell(f(x), y) + \frac{\lambda}{|\mathcal{X}_{weak}|} \sum_{(x,s) \in \mathcal{X}_{weak}} \ell_w(f(x), \boldsymbol{s}). \tag{3}$$

Here, $\ell$ is the cross-entropy loss, $\lambda$ is a loss factor, and $\ell_w$ is a partial label learning loss defined as:

$$\ell_w(f(x), \boldsymbol{s}) = -\sum_{c \in C} \boldsymbol{s}^c \log\left(p(c|x)\right). \tag{4}$$

Following CPL (Zhang et al., 2024), we continue the semi-supervised training over $S$ sessions.

### 3.3 Weakly-supervised sampling

To overcome the limitations of random selection, we introduce a weakly-supervised sampling module that selects the most diverse and representative $N$ samples from the unlabelled set. This module operates through a two-step protocol that can be integrated into any existing semi-supervised method: (***i***) a filtering strategy that leverages the VLM predictions as weak supervision to remove uninformative samples, and (***ii***) a clustering-based selection method to ensure diversity among the selected samples. By leveraging clusters formed based on semantic similarity, our approach ensures diversity in the selected labelled samples, optimizing the use of the limited label budget.

*Step 1: Filtering with weak supervision.* We leverage the zero-shot predictions of the pre-trained VLM as weak supervision to filter the unlabelled set $U$. Specifically, we remove samples with both the highest and lowest confidence predictions by the VLM. Most confident samples offer minimal information gain, as the model is already certain of their classification (refer to section 4.7). Conversely, low-confidence samples are likely to be outliers or noisy data points that can negatively impact model generalization, especially in few-shot learning scenarios where training data is scarce.

For each unlabelled sample $i$, we generate a probability distribution over the output classes with the pre-trained VLM using Eq. 1 as: $\boldsymbol{p}_i = [p_i^1, p_i^2, \cdots, p_i^C]$. We define the confidence score for each sample as the maximum probability value over the classes:

$$c_i = \max_{1 \leq c \leq C} p_i^c = \max\{p_i^1, p_i^2, \cdots, p_i^C\}. \tag{5}$$

We then sort the samples in descending order of confidence: $\mathcal{D}_{\text{sorted}} = \{x_{(1)}, x_{(2)}, \ldots, x_{(N)}\}$, where $x_{(i)}$ is the sample with the $i$-th highest confidence score, satisfying: $c_{(1)} \geq c_{(2)} \geq \ldots \geq c_{(N)}$. Next, we divide the sorted samples into $q$ quantiles, $\{Q_1, Q_2, \cdots Q_q\}$, and remove the first and last quantiles, corresponding to the most and least confident samples. Finally, the filtered unlabelled dataset after the first step can be represented as $\mathcal{D}_{\text{filtered}} = \bigcup_{k=2}^{q-1} Q_k$.

*Step 2: Diversity Sampling.* Next, we select $N$ diverse samples from the filtered dataset $\mathcal{D}_{\text{filtered}}$, with a cluster-based sampling technique. First, we obtain the representations for each sample using a pre-trained vision encoder $\theta$ as $\mathbf{z}_i = \theta(x_i) \in \mathbb{R}^d$, where $\mathbf{z}_i$ is the $d$-dimensional embedding of sample $x_i$. We then

Table 1: Performance across 13 benchmarks on standard semi-supervised tuning with **textual** prompting. Here, SelfPrompt is trained with the same random labelled set as existing methods.

| Methods | Average | Flowers102 | RESISC45 | DTD | CUB | EuroSAT | FGVCAircraft |
|---|---|---|---|---|---|---|---|
| Zero-shot CLIP | 55.17 | $63.67_{0.00}$ | $54.48_{0.00}$ | $43.24_{0.00}$ | $51.82_{0.00}$ | $32.88_{0.00}$ | $17.58_{0.00}$ |
| CoOp | 62.28 | $75.96_{0.74}$ | $68.13_{0.55}$ | $37.10_{5.45}$ | $55.29_{0.59}$ | $62.05_{1.64}$ | $20.02_{0.77}$ |
| GRIP | 67.40 | $83.60_{0.48}$ | $74.11_{0.68}$ | $56.07_{0.79}$ | $56.65_{0.33}$ | $58.66_{2.64}$ | $16.98_{0.20}$ |
| PromptKD | 66.90 | $84.28_{1.77}$ | $77.21_{1.99}$ | $55.19_{1.23}$ | $57.28_{1.78}$ | $59.21_{2.31}$ | $14.17_{1.31}$ |
| CPL | 71.41 | $89.66_{0.36}$ | $80.98_{0.11}$ | $61.21_{0.56}$ | $58.53_{0.24}$ | $77.51_{0.80}$ | $22.48_{0.63}$ |
| **SelfPrompt** | **76.12** | $\mathbf{91.12_{0.31}}$ | $\mathbf{82.19_{0.19}}$ | $\mathbf{70.45_{0.68}}$ | $\mathbf{66.67_{0.18}}$ | $\mathbf{84.14_{0.11}}$ | $\mathbf{33.02_{0.72}}$ |
| Δ | ↑ 4.71 | ↑ 1.46 | ↑ 1.21 | ↑ 9.24 | ↑ 10.14 | ↑ 6.63 | ↑ 10.54 |

| Methods | Caltech101 | MNIST | Food101 | StanfordCars | OxfordPets | SUN397 | UCF101 |
|---|---|---|---|---|---|---|---|
| Zero-shot CLIP | $82.01_{0.00}$ | $25.10_{0.00}$ | $78.81_{0.00}$ | $60.29_{0.00}$ | $84.32_{0.00}$ | $62.54_{0.00}$ | $60.42_{0.00}$ |
| CoOp | $84.69_{1.43}$ | $58.22_{1.98}$ | $76.23_{1.45}$ | $58.23_{2.45}$ | $82.34_{1.44}$ | $62.19_{1.78}$ | $69.19_{1.03}$ |
| GRIP | $85.99_{1.06}$ | $71.78_{2.59}$ | $80.89_{1.14}$ | $62.83_{1.42}$ | $89.40_{0.33}$ | $67.34_{0.98}$ | $71.94_{0.95}$ |
| PromptKD | $84.28_{2.11}$ | $70.24_{2.01}$ | $81.34_{0.99}$ | $64.11_{2.45}$ | $88.28_{1.97}$ | $64.12_{2.56}$ | $70.11_{1.95}$ |
| CPL | $92.87_{1.14}$ | $75.18_{4.40}$ | $79.38_{1.05}$ | $61.93_{1.30}$ | $87.79_{1.31}$ | $66.98_{0.65}$ | $73.88_{1.32}$ |
| **SelfPrompt** | $\mathbf{93.11_{0.91}}$ | $\mathbf{82.12_{0.35}}$ | $\mathbf{81.93_{0.16}}$ | $\mathbf{70.19_{0.32}}$ | $\mathbf{89.64_{0.47}}$ | $\mathbf{69.98_{0.19}}$ | $\mathbf{75.01_{0.43}}$ |
| Δ | ↑ 0.24 | ↑ 6.94 | ↑ 0.59 | ↑ 6.08 | ↑ 0.24 | ↑ 2.64 | ↑ 1.13 |

Table 2: Performance across 13 benchmarks on standard semi-supervised tuning with **visual** prompting. Here, SelfPrompt is trained with the same random labelled set as existing methods.

| Methods | Average | Flowers102 | RESISC45 | DTD | CUB | EuroSAT | FGVCAircraft |
|---|---|---|---|---|---|---|---|
| Zero-shot CLIP | 55.17 | $63.67_{0.00}$ | $54.48_{0.00}$ | $43.24_{0.00}$ | $51.82_{0.00}$ | $32.88_{0.00}$ | $17.58_{0.00}$ |
| VPL | 60.02 | $67.03_{0.65}$ | $65.14_{0.25}$ | $47.60_{1.09}$ | $52.86_{0.42}$ | $52.47_{2.53}$ | $20.14_{0.26}$ |
| GRIP | 64.77 | $67.95_{1.20}$ | $71.22_{0.77}$ | $54.57_{4.86}$ | $53.83_{0.11}$ | $63.48_{3.09}$ | $19.43_{0.50}$ |
| PromptKD | 64.60 | $68.47_{1.35}$ | $70.78_{0.91}$ | $55.12_{4.98}$ | $54.26_{0.23}$ | $64.05_{2.87}$ | $18.89_{0.61}$ |
| CPL | 67.11 | $73.52_{0.37}$ | $78.46_{0.74}$ | $58.74_{0.81}$ | $49.50_{0.42}$ | $72.03_{1.24}$ | $20.51_{0.68}$ |
| **SelfPrompt** | **73.34** | $\mathbf{80.44_{0.51}}$ | $\mathbf{84.12_{0.32}}$ | $\mathbf{69.98_{0.68}}$ | $\mathbf{55.68_{0.28}}$ | $\mathbf{91.53_{0.14}}$ | $\mathbf{21.17_{0.71}}$ |
| Δ | ↑ 6.23 | ↑ 6.92 | ↑ 5.66 | ↑ 11.23 | ↑ 1.85 | ↑ 19.50 | ↑ 0.65 |

| Methods | Caltech101 | MNIST | Food101 | StanfordCars | OxfordPets | SUN397 | UCF101 |
|---|---|---|---|---|---|---|---|
| Zero-shot CLIP | $82.01_{0.00}$ | $25.10_{0.00}$ | $78.81_{0.00}$ | $60.29_{0.00}$ | $84.32_{0.00}$ | $62.54_{0.00}$ | $60.42_{0.00}$ |
| VPL | $84.29_{1.51}$ | $42.53_{14.1}$ | $78.85_{1.11}$ | $61.25_{2.01}$ | $84.78_{1.01}$ | $62.01_{1.86}$ | $61.25_{1.11}$ |
| GRIP | $87.45_{1.21}$ | $69.66_{5.51}$ | $79.15_{1.32}$ | $61.01_{1.12}$ | $85.44_{0.39}$ | $63.56_{0.90}$ | $65.27_{0.99}$ |
| PromptKD | $86.72_{2.34}$ | $68.19_{4.87}$ | $78.43_{1.56}$ | $62.27_{1.67}$ | $84.13_{0.58}$ | $62.11_{1.12}$ | $66.45_{1.03}$ |
| CPL | $91.03_{1.03}$ | $71.23_{2.67}$ | $77.19_{1.19}$ | $62.01_{1.04}$ | $86.34_{1.51}$ | $64.58_{0.79}$ | $67.28_{1.52}$ |
| **SelfPrompt** | $\mathbf{94.96_{0.91}}$ | $\mathbf{79.71_{0.47}}$ | $\mathbf{79.50_{0.30}}$ | $\mathbf{64.01_{0.30}}$ | $\mathbf{90.11_{0.47}}$ | $\mathbf{68.15_{0.18}}$ | $\mathbf{74.09_{0.43}}$ |
| Δ | ↑ 3.93 | ↑ 8.48 | ↑ 2.31 | ↑ 1.75 | ↑ 3.77 | ↑ 3.57 | ↑ 6.81 |

apply $k$-means clustering to group the samples into $N$ clusters $\mathcal{C} = \{\mathcal{C}_1, \mathcal{C}_2, \ldots, \mathcal{C}_N\}$, such that each cluster contains semantically similar samples, while different clusters have diverse semantics. For each cluster, $j \in \{1, 2, ..., N\}$, we select the sample closest to the cluster center:

$$x_j^* = \arg\min_{x_i \in \mathcal{C}_j} \|\mathbf{z}_i - \boldsymbol{\mu}_j\|^2, \tag{6}$$

where $\boldsymbol{\mu}_j$ is the center of cluster $\mathcal{C}_j$, and $x_j^*$ is the selected sample. Finally, our labelled set is formed by gathering the labels of the selected samples, $\mathcal{X}_L = \{(x_1^*, y_1), (x_2^*, y_2), \cdots, (x_N^*, y_N)\}$. Our weakly-supervised sample technique is illustrated in Figure 3 (left).

## 4 Experiments

### 4.1 Experiment Setup

We report the results of our experiments on 13 datasets of diverse semantic concepts. For evaluating the proposed solution, we closely follow the training and evaluation protocol established by Zhang et al. (2024) and Menghini et al. (2023). Specifically, we utilize few samples from the training set of a dataset as the labelled set and the remaining samples as the unlabelled set. All experiments are conducted in a 2-shot setup, with ten sessions of iterative pseudo-labelling (50 epochs/session). The model is optimized using SGD with a learning rate of 0.02 and a batch size of 64. Main results are reported with $q = 5, \tau = 0.05$, and $\lambda = 1$. Results are reported as the average accuracy and the standard deviation over three runs with random seeds.

Table 3: Performance across 13 benchmarks on active semi-supervised tuning with textual prompting. We have reproduced the results for previous SOTA methods on this setup.

| Methods | Average | Flowers102 | RESISC45 | DTD | CUB | EuroSAT | FGVCAircraft |
|---|---|---|---|---|---|---|---|
| GRIP | 69.12 | $86.59_{0.48}$ | $76.88_{0.68}$ | $58.70_{0.79}$ | $58.19_{0.33}$ | $60.08_{2.64}$ | $17.45_{0.20}$ |
| CPL | 73.08 | $91.12_{0.36}$ | $82.19_{0.11}$ | $63.45_{0.56}$ | $60.67_{0.24}$ | $78.14_{0.80}$ | $22.89_{0.63}$ |
| **SelfPrompt** | **79.33** | $\mathbf{93.04_{0.33}}$ | $\mathbf{85.58_{0.18}}$ | $\mathbf{72.18_{0.78}}$ | $\mathbf{68.84_{0.16}}$ | $\mathbf{87.49_{0.12}}$ | $\mathbf{36.71_{0.70}}$ |
| Δ | ↑ 6.25 | ↑ 1.92 | ↑ 3.39 | ↑ 8.73 | ↑ 8.17 | ↑ 9.35 | ↑ 13.82 |

| | Caltech101 | MNIST | Food101 | StanfordCars | OxfordPets | SUN397 | UCF101 |
|---|---|---|---|---|---|---|---|
| GRIP | $87.64_{1.06}$ | $74.45_{2.59}$ | $81.09_{1.14}$ | $64.43_{1.42}$ | $89.52_{0.33}$ | $70.28_{0.98}$ | $73.31_{0.95}$ |
| CPL | $92.98_{1.14}$ | $76.86_{4.40}$ | $81.93_{1.05}$ | $65.19_{1.30}$ | $89.64_{1.31}$ | $69.98_{0.65}$ | $75.01_{1.32}$ |
| **SelfPrompt** | $\mathbf{94.10_{0.92}}$ | $\mathbf{90.23_{0.36}}$ | $\mathbf{82.19_{0.17}}$ | $\mathbf{75.21_{0.33}}$ | $\mathbf{89.86_{0.48}}$ | $\mathbf{74.77_{0.18}}$ | $\mathbf{81.07_{0.44}}$ |
| Δ | ↑ 1.12 | ↑ 13.37 | ↑ 0.26 | ↑ 10.02 | ↑ 0.22 | ↑ 4.49 | ↑ 6.06 |

Table 4: Performance across 13 benchmarks on semi-supervised tuning with **textual** prompting with varying shots.

| Setting | Methods | Average | Flowers102 | RESISC45 | DTD | CUB | EuroSAT | Aircraft | Caltech101 | MNIST | Food101 | Cars | OxfordPets | SUN397 | UCF101 |
|---|---|---|---|---|---|---|---|---|---|---|---|---|---|---|---|
| 1-shot | CPL | 66.69 | 89.13 | 73.44 | 48.67 | 52.13 | 74.63 | 19.03 | 92.22 | 58.62 | 78.65 | 58.98 | 85.51 | 65.43 | 70.60 |
| | SelfPrompt | **78.48** | **92.47** | **84.09** | **71.49** | **69.57** | **83.33** | **36.29** | **94.74** | **84.97** | **82.15** | **75.81** | **89.69** | **74.63** | **81.06** |
| 2-shot | CPL | 71.41 | 89.66 | 80.98 | 61.21 | 58.53 | 77.51 | 22.48 | 92.87 | 75.18 | 79.38 | 61.93 | 87.79 | 66.98 | 73.88 |
| | SelfPrompt | **79.33** | **93.04** | **85.58** | **72.18** | **68.84** | **87.49** | **36.71** | **94.10** | **90.23** | **82.19** | **75.21** | **89.86** | **74.77** | **81.07** |
| 4-shot | CPL | 68.42 | 89.88 | 72.84 | 53.34 | 52.77 | 75.96 | 19.08 | 93.38 | 64.83 | 78.85 | 66.23 | 86.28 | 65.45 | 70.62 |
| | SelfPrompt | **79.48** | **93.14** | **85.60** | **72.19** | **69.62** | **87.44** | **36.75** | **94.75** | **90.25** | **82.36** | **75.36** | **89.90** | **74.79** | **81.15** |
| 8-shot | CPL | 70.08 | 89.90 | 74.93 | 58.46 | 56.33 | 76.21 | 21.34 | 93.44 | 70.24 | 79.01 | 67.83 | 86.35 | 66.01 | 70.95 |
| | SelfPrompt | **80.06** | **93.64** | **86.20** | **72.59** | **70.32** | **87.99** | **37.40** | **95.20** | **90.85** | **82.86** | **76.06** | **90.45** | **75.39** | **81.80** |
| 16-shot | CPL | 70.66 | 90.50 | 75.43 | 59.16 | 56.88 | 76.86 | 21.79 | 93.94 | 70.84 | 79.71 | 68.38 | 86.95 | 66.66 | 71.45 |
| | SelfPrompt | **80.64** | **94.19** | **86.80** | **73.09** | **70.97** | **88.69** | **37.85** | **95.70** | **91.45** | **83.41** | **76.76** | **91.10** | **75.89** | **82.40** |

**Datasets.** Following Zhang et al. (2024) and Menghini et al. (2023) we use Flower102 (Nilsback & Zisserman, 2008), Resisc-45 (Cheng et al., 2017), DTD (Cimpoi et al., 2014), CUB-200 (Wah et al., 2011), EuroSAT (Helber et al., 2019), FGVCAircraft (Maji et al., 2013), and MNIST (Deng, 2012). Additionally, we use the following 6 datasets, bringing the total to **13 datasets**: Caltech101 (Fei-Fei et al., 2004), Food101 (Bossard et al., 2014), StanfordCars (Krause et al., 2013), SUN397 (Xiao et al., 2010), OxfordPets (Parkhi et al., 2012), UCF101 (Soomro et al., 2012).

**Protocol.** For evaluating the proposed solution, we closely follow the training and evaluation protocol established by Zhang et al. (2024) and Menghini et al. (2023). Specifically, we utilize a few samples from the training set of a dataset as the labelled set and the remaining samples as the unlabelled set, followed by the assessment of the trained model on the test set.

**Implementation details.** Following Zhang et al. (2024) and Menghini et al. (2023) we adopt a CLIP ViT-B/32 (Radford et al., 2021) as the pre-trained backbone of our model. All experiments are conducted in a 2-shot setup, with ten sessions of iterative pseudo-labelling (50 epochs per session). The model is optimized using SGD with a learning rate of 0.02 and a batch size of 64. The main results are reported with the value of $q = 5, \tau = 0.05$, and $\lambda = 1$. Results are reported as the average accuracy and the standard deviation over three runs with random seeds. Training is performed on a single Nvidia V100 GPU.

## 4.2 Standard Semi-supervised learning

First, we present our results on standard semi-supervised learning in a 2-shot setup. The results of this experiment are presented in Table 1, where we observe that SelfPrompt shows large and consistent improvements over prior works across the 13 datasets. On average, SelfPrompt achieves an accuracy of 76.12% with just two labelled samples per class, which is a 4.71% improvement over the previous SOTA CPL. Notably, SelfPrompt shows up to 10.54% improvement over the previous SOTA on individual datasets. More importantly, SelfPrompt shows higher improvements on datasets with lower zero-shot (VLM) accuracies (e.g., FGVCAircraft and MNIST). We also report the results for visual prompt tuning in Table 2. Similar to textual prompting, SelfPrompt with visual prompt tuning outperforms CPL by up to 19.50%, with a 6.23%,

Table 5: Impact of different encoders.

| Encoder | Parameters | Accuracy |
|---|---|---|
| CLIP-B/32 | 151.3 M | 79.33 |
| CLIP-B/16 | 149.6 M | 82.03 |
| CLIP-L/14 | 427.6 M | 86.54 |
| MetaClip-B/16 | 151.3 M | 80.15 |

which is higher than the improvement with text prompts. On average, SelfPrompt with visual prompt tuning achieves an average accuracy of 73.34%.

### 4.3 Active Semi-supervised Learning

Next, we evaluate the performance of existing SOTA in active semi-supervised learning settings with our weakly-supervised sampling module. As we find from Table 3, all existing methods show improved performance with our weakly-supervised sampling module (compared to Table 1). Nonetheless, SelfPrompt consistently outperforms priors SOTA across the datasets. On average, SelfPrompt shows 6.25% improvement over existing methods, with up to 13.82% improvements on individual datasets.

**Performance on different shots.** To further investigate the effectiveness of SelfPrompt in label-scarce scenarios, we evaluate its performance under various few-shot settings, specifically 1-, 2-, and 4-shot configurations. The results, presented in Table 4, demonstrate SelfPrompt's consistent superiority over the previous SOTA across all few-shot settings. Notably, our performance improvement is more pronounced with fewer labelled samples. For instance, our method achieves an average accuracy improvement of 7.92% with 2-shots, compared to an 11.78% improvement with a 1-shot setup. Interestingly, in a few cases, SelfPrompt with 1-shot performs slightly better than 2-shots. This may occur because increasing the number of labelled samples can in turn increase the number of initial pseudo-labels selected by our cluster-guided pseudo-labelling module ($\mathcal{X}^+$). As we select $p$ samples per cluster, the likelihood of including incorrect pseudo-labels also increases. While this issue can be resolved with a dataset-specific hyperparameter for this module, we opt to not resort to such approaches for more generalizability. Another interesting observation is that CPL's performance, unlike ours, does not improve with additional labelled samples (4-shot) compared to the 2-shot setup, and even degrades by 2.99%, further highlighting the effectiveness of our method.

**Performance of pseudo-labelling.** In this section, we evaluate the performance of pseudo-labelling in our proposed SelfPrompt method. As shown earlier in Figure 1 (left), SelfPrompt achieves significantly higher pseudo-label accuracy compared to previous methods. The plots are calculated on the FGVCAircraft dataset, which is the most difficult dataset for the VLM, which is evident from the low zero-shot accuracy of the VLM. SelfPrompt achieves higher pseudo-label accuracy from the very first session, driven by the labelled set selection module, which selects the most representative samples as the labelled set, and the cluster-guided pseudo-labelling module, which ensures high-quality pseudo-labels. As the training progresses, after iteration 3, CPL shows a drop in the pseudo-label accuracy. This results in a sub-optimal test accuracy after iteration 4 (see Figure 1 (right). In contrast, SelfPrompt gradually improves the pseudo-label and test accuracy.

**Performance on different backbones.** Previously, we presented our main results using the CLIP-B/32 backbone, following (Menghini et al., 2023; Zhang et al., 2024). In this section, we further evaluate the versatility of SelfPrompt by exploring its performance across various encoder architectures as backbones. Specifically, we evaluate CLIP-B/32, CLIP-B/16, and CLIP-L/14 as encoders for our method. As shown in Table 5, SelfPrompt exhibits strong generalizability across different encoder sizes, including the large-scale CLIP-L/14 with 427.6 million parameters. Utilizing CLIP-L/14 as the backbone achieves an accuracy of 86.54%, representing a 7.21% improvement over the default encoder.

### 4.4 Base-to-novel generalization

Base-to-novel generalization is a widely adopted evaluation protocol in the few-shot tuning literature (Zhou et al., 2022a). This setup involves fine-tuning the VLM on a subset of classes with a few labelled samples and evaluating its performance on the seen classes, as well as its zero-shot performance on the unseen classes. We evaluate our proposed solution on the evaluation protocol of PromptKD (Li et al., 2024) by

Table 6: Comparison with existing methods on base-to-novel generalization in a **2-shot training** setup. The notation "ul." indicates that the method is trained using both the unlabelled set and the labelled set.

(a) Average.

| ViT-B/16 | ul. | Base | Novel | HM |
|---|---|---|---|---|
| CLIP | ✗ | 69.3 | 74.2 | 71.7 |
| Co-CoOp | ✗ | 71.9 | 73.4 | 72.6 |
| MaPLe | ✗ | 74.9 | 73.3 | 74.0 |
| PromptSRC | ✗ | 78.1 | 74.7 | 76.3 |
| CoPrompt | ✗ | 74.2 | 72.4 | 73.1 |
| PromptKD | ✓ | 79.7 | 76.8 | 78.1 |
| SelfPrompt | ✓ | **85.6** | **80.8** | **83.0** |
| Δ | | ↑ 5.9 | ↑ 4.0 | ↑ 4.9 |

(b) ImageNet

| ViT-B/16 | ul. | Base | Novel | HM |
|---|---|---|---|---|
| CLIP | ✗ | 72.4 | 68.1 | 70.2 |
| Co-CoOp | ✗ | 72.5 | 69.1 | 70.8 |
| MaPLe | ✗ | 75.5 | 70.5 | 72.9 |
| PromptSRC | ✗ | 75.2 | 69.8 | 72.4 |
| CoPrompt | ✗ | 74.7 | 70.9 | 72.8 |
| PromptKD | ✓ | 75.3 | 70.8 | 73.0 |
| SelfPrompt | ✓ | **75.6** | **71.3** | **73.4** |
| Δ | | ↑ 0.1 | ↑ 0.4 | ↑ 0.4 |

(c) Caltech101

| ViT-B/16 | ul. | Base | Novel | HM |
|---|---|---|---|---|
| CLIP | ✗ | 96.8 | 94.0 | 95.4 |
| Co-CoOp | ✗ | 93.7 | 93.3 | 93.5 |
| MaPLe | ✗ | 96.3 | 96.2 | 96.2 |
| PromptSRC | ✗ | 97.0 | 94.4 | 95.7 |
| CoPrompt | ✗ | 97.7 | 95.6 | 96.6 |
| PromptKD | ✓ | 98.1 | **96.5** | 97.3 |
| SelfPrompt | ✓ | **99.1** | 96.3 | **97.7** |
| Δ | | ↑ 0.4 | ↓ 0.2 | ↑ 0.4 |

(d) OxfordPets

| ViT-B/16 | ul. | Base | Novel | HM |
|---|---|---|---|---|
| CLIP | ✗ | 91.1 | 97.2 | 94.1 |
| Co-CoOp | ✗ | 93.3 | 97.8 | 95.5 |
| MaPLe | ✗ | 90.1 | 91.1 | 90.6 |
| PromptSRC | ✗ | 94.9 | 96.8 | 95.8 |
| CoPrompt | ✗ | 92.0 | 96.8 | 94.3 |
| PromptKD | ✓ | 96.1 | 97.1 | 96.6 |
| SelfPrompt | ✓ | **96.3** | **98.1** | **97.2** |
| Δ | | ↑ 0.2 | ↑ 0.3 | ↑ 0.6 |

(e) StanfordCars

| ViT-B/16 | ul. | Base | Novel | HM |
|---|---|---|---|---|
| CLIP | ✗ | 63.3 | 74.8 | 68.6 |
| Co-CoOp | ✗ | 64.5 | 73.4 | 68.7 |
| MaPLe | ✗ | 67.8 | 74.5 | 71.0 |
| PromptSRC | ✗ | 67.8 | 74.2 | 70.9 |
| CoPrompt | ✗ | 64.9 | 72.9 | 68.7 |
| PromptKD | ✓ | 77.1 | 84.1 | 80.4 |
| SelfPrompt | ✓ | **80.4** | **84.2** | **82.3** |
| Δ | | ↑ 3.3 | ↑ 0.1 | ↑ 1.9 |

(f) Flowers102

| ViT-B/16 | ul. | Base | Novel | HM |
|---|---|---|---|---|
| CLIP | ✗ | 72.0 | **77.8** | 74.8 |
| Co-CoOp | ✗ | 78.6 | 76.0 | 77.3 |
| MaPLe | ✗ | 85.9 | 75.8 | 80.5 |
| PromptSRC | ✗ | 89.9 | 77.8 | 83.4 |
| CoPrompt | ✗ | 80.0 | 72.8 | 76.2 |
| PromptKD | ✓ | 87.2 | 73.5 | 79.8 |
| SelfPrompt | ✓ | **99.1** | 81.6 | **89.5** |
| Δ | | ↑ 9.2 | ↑ 3.8 | ↑ 6.1 |

(g) Food101

| ViT-B/16 | ul. | Base | Novel | HM |
|---|---|---|---|---|
| CLIP | ✗ | 90.1 | 91.2 | 90.6 |
| Co-CoOp | ✗ | 90.5 | 91.3 | 90.9 |
| MaPLe | ✗ | 89.6 | 90.1 | 89.8 |
| PromptSRC | ✗ | 89.2 | 91.2 | 90.2 |
| CoPrompt | ✗ | 87.2 | 91.3 | 89.2 |
| PromptKD | ✓ | 90.9 | 92.7 | 91.8 |
| SelfPrompt | ✓ | **92.4** | **93.6** | **93.0** |
| Δ | | ↑ 1.5 | ↑ 0.9 | ↑ 1.2 |

(h) FGVCAircraft

| ViT-B/16 | ul. | Base | Novel | HM |
|---|---|---|---|---|
| CLIP | ✗ | 27.2 | 36.3 | 31.1 |
| Co-CoOp | ✗ | 30.3 | 37.1 | 33.4 |
| MaPLe | ✗ | 32.9 | 35.8 | 34.3 |
| PromptSRC | ✗ | 33.1 | 33.3 | 33.2 |
| CoPrompt | ✗ | 26.0 | 17.1 | 20.6 |
| PromptKD | ✓ | 16.8 | 12.3 | 14.2 |
| SelfPrompt | ✓ | **44.5** | **43.2** | **43.8** |
| Δ | | ↑ 11.4 | ↑ 6.1 | ↑ 9.5 |

(i) SUN397

| ViT-B/16 | ul. | Base | Novel | HM |
|---|---|---|---|---|
| CLIP | ✗ | 69.4 | 75.4 | 72.2 |
| Co-CoOp | ✗ | 73.3 | 77.8 | 75.5 |
| MaPLe | ✗ | 75.6 | 76.3 | 75.9 |
| PromptSRC | ✗ | 78.0 | 77.2 | 77.6 |
| CoPrompt | ✗ | 78.7 | 78.7 | 78.7 |
| PromptKD | ✓ | 80.2 | 80.8 | 80.5 |
| SelfPrompt | ✓ | **84.3** | **81.6** | **82.9** |
| Δ | | ↑ 4.1 | ↑ 0.8 | ↑ 2.4 |

(j) DTD

| ViT-B/16 | ul. | Base | Novel | HM |
|---|---|---|---|---|
| CLIP | ✗ | 53.2 | 59.9 | 56.4 |
| Co-CoOp | ✗ | 61.1 | 53.4 | 57.0 |
| MaPLe | ✗ | 63.7 | 60.4 | 62.0 |
| PromptSRC | ✗ | 71.9 | 55.3 | 62.5 |
| CoPrompt | ✗ | 65.2 | 60.5 | 62.8 |
| PromptKD | ✓ | 80.2 | **67.0** | 73.0 |
| SelfPrompt | ✓ | **86.0** | 64.3 | **73.6** |
| Δ | | ↑ 5.8 | ↓ 2.7 | ↑ 0.6 |

(k) EuroSAT

| ViT-B/16 | ul. | Base | Novel | HM |
|---|---|---|---|---|
| CLIP | ✗ | 56.5 | 64.1 | 60.0 |
| Co-CoOp | ✗ | 59.1 | 66.8 | 62.7 |
| MaPLe | ✗ | 68.2 | 60.6 | 64.2 |
| PromptSRC | ✗ | 80.6 | 75.5 | 78.0 |
| CoPrompt | ✗ | 68.8 | 60.1 | 64.2 |
| PromptKD | ✓ | 88.4 | 89.1 | 88.7 |
| SelfPrompt | ✓ | **96.4** | **93.3** | **94.8** |
| Δ | | ↑ 8.0 | ↑ 4.2 | ↑ 6.1 |

(l) UCF101

| ViT-B/16 | ul. | Base | Novel | HM |
|---|---|---|---|---|
| CLIP | ✗ | 70.5 | 77.5 | 73.9 |
| Co-CoOp | ✗ | 74.2 | 71.9 | 73.0 |
| MaPLe | ✗ | 78.6 | 75.5 | 77.0 |
| PromptSRC | ✗ | 81.9 | 77.0 | 79.4 |
| CoPrompt | ✗ | 81.1 | 79.3 | 80.2 |
| PromptKD | ✓ | 86.3 | 80.4 | 83.2 |
| SelfPrompt | ✓ | **87.8** | **81.1** | **84.3** |
| Δ | | ↑ 1.5 | ↑ 0.7 | ↑ 1.1 |

Table 7: Ablation study. W.S.S., C.G.P., and C.A.SSL correspond to weakly-sup sampling, cluster-guided pseudo-labelling, and confidence-aware SSL.

| W.S.S. | C.G.P. | C.A.SSL | Accuracy |
|---|---|---|---|
| ✓ | ✓ | ✓ | 79.33 |
| ✗ | ✓ | ✓ | 76.12 |
| ✓ | ✗ | ✓ | 74.39 |
| ✓ | ✓ | ✗ | 78.01 |
| ✗ | ✗ | ✓ | 73.49 |
| ✗ | ✓ | ✗ | 75.67 |
| ✓ | ✗ | ✗ | 73.08 |
| ✗ | ✗ | ✗ | 71.41 |

utilizing the unlabelled data along with the labelled data. Table 6 presents the results of our experiments and their comparison to prior works in a **2-shot** evaluation setup. As we observe in this table, SelfPrompt shows an average improvement of 4.9% in the harmonic mean over the prior works. SelfPrompt not only improves the performance of the seen classes but also improves the generalization of the unseen (novel) classes. Specifically, the improvement on the base and novel classes are 5.9% and 4.0% on average. More importantly, SelfPrompt shows greater improvements on the datasets where prior works, as well as the pre-trained VLM, show suboptimal performances. For example, SelfPrompt shows a 9.5% improvement on the FGVCAircraft dataset, where the accuracy of the pre-trained CLIP is only 31.1%.

## 4.5 Ablation

We present an ablation study on our proposed method in Table 9. Here, W.S.S., C.G.P., and C.A.SSL correspond to the three modules of our proposed solution, namely, weakly-supervised sampling, cluster-guided pseudo-labelling, and confidence-aware semi-supervised learning. The results are reported as the average accuracy over all datasets. Here, our model has an average accuracy of 79.33% (also shown in Table 1), while removing all three components results in an accuracy of 71.41%. As we find from this table, cluster-

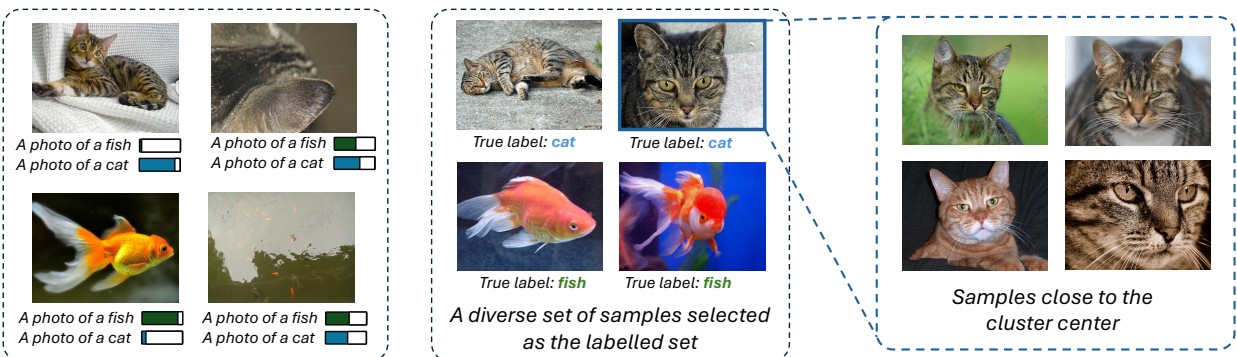

Figure 4: Qualitative analysis of weakly-supervised sampling and cluster-guided pseudo-labelling with two classes (fist and cat). (left) Illustrations of the most confident samples, which provide minimal information gain, alongside the least confident samples, which are less representative of their respective classes. (middle) Examples of selected samples demonstrating high semantic diversity. (right) Samples close to the cluster centers) exhibit high visual and semantic similarity.

guided pseudo-labelling has a high impact on the overall performance, removing which results in a 4.94% drop in the performance. This is also evident from the fact that including only this component shows a 4.26% improvement over the baseline. Next, removing weakly-supervised sampling shows a 3.21% drop in performance, while removing confidence-aware semi-supervised learning shows a 1.32% drop in performance. Another notable observation is that including weakly-supervised sampling alone provides a 1.61% improvement, but including it with cluster-guided pseudo-labelling shows an additional 2.34% improvement. These findings suggest that selecting a more representative set of labelled samples also improves the pseudo-label quality of the clutter-guided pseudo-labelling module.

## 4.6 Computational complexity

In this section, we discuss the computational complexity of our proposed solution. Since we do not change the model architecture or the parameters, our model is as efficient as prior works such as (Zhang et al., 2024) during inference. In the training stage, our proposed solution introduces some additional computations at the beginning of the training due to the introduction of sample selection and the pseudo-labelling strategy. Here, the algorithm requires the representations for all the unlabelled samples. The computational cost of this step is equivalent to a forward pass over all the unlabelled samples with the pre-trained encoder, which adds a negligible computational cost compared to the overall training process. The clustering process is also a small one-time operation at the beginning of the training. Overall, the training time of SelfPrompt is on par with the previous SOTA CPL (see Table 8e for training time on EuroSAT).

## 4.7 Discussion

The primary reason for excluding both high-confidence and low-confidence samples is that retaining high-confidence samples (in the labelled set) results in representations that fail to generalize effectively to distributions outside the selected labelled set. Previous works, such as (Roy & Etemad, 2024; Sarkar et al., 2024) have demonstrated that overconfidence on a specific distribution hinders effective generalization. This issue is particularly pronounced when the model has not been fine-tuned on the distribution of the specific downstream task. Accordingly, we hypothesize that selecting a more diverse and representative set of samples and excluding both high- and low-confidence samples would improve generalization. To validate this hypothesis, we conduct experiments comparing the outcomes of different sampling strategies: (a) removing only the low-confidence samples, (b) removing only the high-confidence samples, (c) keeping only the high-confidence samples, and (d) keeping only the low-confidence samples. The results averaged across all datasets with $q = 5$, are presented below. These experiments were performed using only the weakly-supervised sampling module to isolate the behaviour of this specific module. As we find from Table 9, excluding both the high- and low-confidence samples yields the best performance.

Table 8: Sensitivity analysis of different components of SelfPrompt, averaged across 13 datasets.

| (a) **Cluster algo.** | | (b) **Filtering thresh.** | | (c) **Cluster samples** | | (d) **Sup. samples** | | (e) **Time complexity** | |
|---|---|---|---|---|---|---|---|---|---|
| Method | Accuracy | $q$ | Accuracy | $p$ | Accuracy | $\tau$ | Accuracy | Method | Time (H) |
| $k$-means | 79.33 | 3 | 78.25 | 5 | 77.01 | 0.05 | 79.33 | CPL | 2:04 |
| $k$-means++ | 79.35 | 5 | 79.33 | 20 | 79.03 | 0.10 | 79.10 | SelfPrompt | 2:10 |
| Bi. $k$m++ | 79.29 | 10 | 79.12 | 50 | 79.33 | 0.20 | 79.01 | | |
| | | 20 | 78.75 | 75 | 79.31 | | | | |

Table 9: Ablation study. W.S.S., C.G.P., and C.A.SSL correspond to weakly-sup sampling, cluster-guided pseudo-labelling, and confidence-aware SSL.

| Setting | Accuracy |
|---|---|
| Ours (removes both high- and low-confidence samples) | **73.08** |
| Removing only the low-confidence samples | 72.03 |
| Removing only the high-confidence samples | 72.43 |
| Keeping only the high-confidence samples | 71.78 |
| Keeping only the low-confidence samples | 72.55 |

**Qualitative analysis.** We present a qualitative analysis of our weakly supervised sampling module and cluster-guided pseudo-labelling module in Figure 4. As shown in Figure 4 (left), most confident samples distinctly represent their corresponding classes, and thus may not provide additional information to fine-tune the VLM. In contrast, low-confidence samples lack a clear visual representation of the desired class objects, and therefore their inclusion may not effectively aid the model's learning process. Figure 4 (middle) depicts the diverse set of samples selected by our model. Finally, Figure 4 (right) shows one example of clutter-guided pseudo-labelling, where samples close to the clutter centers are visually and semantically similar to the labelled sample, and can be assigned to the same pseudo-labels.

## 4.8 Sensitivity Study

Next, we present a comprehensive set of experiments on different components of our method. First, we study the performance of different clustering methods for the weakly-supervised sampling module. To this end, we explore $k$-means, $k$-means++, and Bisecting $k$-means++ as the clustering method. The results of this study are presented in Table 8a. We find from this study that SelfPrompt is not very sensitive to the choice of clustering algorithm, as all three clustering methods show similar performances, with the best accuracy being 79.35%. Next, Table 8b presents a study on filtering using different confidence intervals, using different quantiles $q = 3$, 5, and 20, during step 1 of our weakly supervised sampling module. Here, $q = 20$ indicates that the unlabelled dataset is divided into 20 quantiles and removes the lowest and highest 5% of samples while retaining 90% of the samples. Here, we observe that the best result is obtained when we filter the most and least confident 20% of samples ($q = 5$), indicating that discarding of more low-confidence samples improves performance. Next, we study the impact of the number of pseudo-labels ($p$) selected with the cluster-guided pseudo-labelling approach. Specifically, we study the performance of selecting 5, 20, and 50 samples per cluster. The results are presented in Table 8c, where we observe that performance improves as the number of samples per cluster increases. We specifically observe a significant improvement when increasing the number of samples from 5 to 20, with the best performance achieved at 50 samples per cluster. Finally, we study the performance of selecting different portions of most confident samples ($\tau$) as the labelled set. The results are presented in Table 8d, where the best performance is achieved when selecting the most confident 5% of the sample ($\tau = 0.05$) as pseudo-labels.

## 5 Conclusion

In this paper, we propose SelfPrompt, a novel prompt-tuning approach for vision-language models in semi-supervised setups. Our method addresses three key limitations of prior works: under-utilization of the limited label-set budget, reliance on miscalibrated VLMs for pseudo-labelling, and the accumulation of noisy pseudo-labels. We demonstrate SelfPrompt's superior performance across 13 datasets, in standard semi-supervised learning, active semi-supervised learning, and base-to-novel generalization tasks. These findings pave the

way for more effective and scalable utilization of vision-language models in diverse semi-supervised learning scenarios. A limitation of our proposed solution is that it is slightly more computationally expensive.

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
