# OpenReview forum: "SelfPrompt: Confidence-Aware Semi-Supervised Tuning for Improved Vision-Language Model Adaptation"
_TMLR — Accepted by TMLR_

### Review · Reviewer_TEbE · 2026-02-15

**Summary Of Contributions:**

The paper proposes SelfPrompt, a confidence-aware semi-supervised method for adapting pre-trained vision-language models with few labeled examples by reducing pseudo-label noise and improving labeled data selection. The paper addresses noisy pseudo-label accumulation under domain shift and miscalibration by using cluster-guided pseudo-labeling rather than directly trusting VLM predictions and also improves label efficiency under a limited labeling budget by filtering out the most and least confident samples then running k-means on image features to select diverse representative samples.

**Additional Comments:**

N/A

**Audience:**

Yes

**Audience Explanation:**

The paper should interest part of TMLR’s audience because it addresses a practical failure mode in semi-supervised adaptation of vision-language models under domain shift and miscalibration, and it provides a clear, modular method with empirical results and ablations.

**Claims And Evidence:**

Yes

**Claims Explanation:**

Overall, the paper’s claims are supported by clear experimental results with appropriate baselines, and the reported gains are reinforced by ablations and sensitivity studies that help attribute improvements to specific components. However, addressing the following concerns would further strengthen the paper and make the evidence more convincing.

W1. The paper would add detailed explanations around Figure 1 and the experimental protocol, including a clearer justification of why the compared methods are particularly limited under this exact setting and what aspects of the setting drive those limitations.

W2. The confidence filtering step that removes the most and least confident samples is plausible in standard benchmarks, but it is unclear how reliably this assumption holds in specialized high-shift domains such as medical imaging where miscalibration and distribution shift can be much more severe.

W3. Since the diversity sampling relies on k-means over image features, it would be useful to discuss or test whether simple guided clustering variants could further improve robustness, for example incorporating text-side information or external guidance as explored in related directions [1, 2].

[1] Li et al., Image Clustering with External Guidance, ICML 25.

[2] Kim et al., Active Prompt Learning with Vision-Language Model Priors, TMLR 25.

W4. The evaluation emphasizes the 2-shot regime where the method is particularly effective, but it would strengthen the evidence to include more conventional label budgets such as 16-shot and to test additional VLM backbones to confirm the gains are not overly specific to a particular low-shot setting or model choice.

**Requested Changes:**

I recommend addressing the weaknesses noted above.

---

> ### Author Response · Authors · 2026-03-25
>
> > Clarity of Figure 1 and experimental protocol motivation
>
> We thank the reviewer for all their comments and feedback.
> Figure 1 presents the standard training setup on a 2-shot setup, where training runs over sessions. This figure illustrates the problem statement discussed in the second paragraph of the introduction, namely, the impact of initial and continuous accumulation of noisy pseudo-labels on overall performance.
>
> > Reliability of confidence filtering in high-shift domains
>
> High domain shift does not significantly affect the method in practice, as the filtering is quantile-based thresholding rather than absolute. The fundamental assumption that highly confident samples are too easy and very low-confidence samples are noisy holds irrespective of the domain.
>
>
>
> > Exploring external guidance for clustering
>
>
> Thank you for this interesting idea. However, exploring such external guidance will make any comparison to existing methods unfair. In this work, we focus on improving the performance while having identical supervision and experimental setup as those established in the literature. We leave this direction for future work to explore.

---

### Review · Reviewer_EiFH · 2026-03-01

**Summary Of Contributions:**

This paper investigates how to improve the performance of vision-language models (VLM) on downstream tasks using semi-supervised prompt-tuning. The authors identify three primary limitations in previous VLM tuning methods: the under-utilization of limited labeled sets, the miscalibration of VLM predictions, and the accumulation of noisy pseudo-labels. To address these challenges, they propose a framework called SelfPrompt, which consists of three core components: Cluster-Guided Pseudo-Labeling (C.G.P.), Confidence-Aware Semi-Supervised Learning (C.A.SSL), and Weakly-Supervised Sampling (W.S.S.). The authors evaluate SelfPrompt across 13 datasets and demonstrate that it surpasses existing state-of-the-art methods across various downstream tasks.

**Additional Comments:**

N/A

**Audience:**

Yes

**Audience Explanation:**

The submission demonstrates several strengths that would be of interest to TMLR's audience:

- Technically sound motivation: The paper is motivated by a thorough analysis of the limitations inherent in prior VLM tuning methods, specifically addressing under-utilization of label budgets, model miscalibration, and the accumulation of noisy pseudo-labels . SelfPrompt provides a novel, systematic integration of components—C.G.P., C.A.SSL, and W.S.S.—that are specifically tailored to resolve these VLM-centric challenges.

- Comprehensive empirical validation: The authors provide an extensive evaluation across 13 diverse datasets, doubling the scope of prior works like CPL and GRIP, which typically utilize 6 datasets. Furthermore, the study explores three distinct downstream tuning settings—standard semi-supervised, active semi-supervised, and base-to-novel generalization—providing a broad perspective on the framework's utility.

- Superior low-shot performance: SelfPrompt consistently outperforms existing state-of-the-art methods, with particularly pronounced gains in extremely label-scarce environments. As shown in Table 4, the performance margin is most significant in 1-shot and 2-shot settings, where it exceeds previous benchmarks by a substantial margin (e.g., achieving over an 11% improvement in the 1-shot setup).

**Broader Impact Concerns:**

None.

**Claims And Evidence:**

Yes

**Claims Explanation:**

- Novelty: The novelty claim is generally supported. SelfPrompt is a novel semi-supervised prompt-tuning approach for VLMs, justified by its in-depth analysis of the limitations in prior methods—such as miscalibration and noise accumulation—and its systematic integration of complementary components tailored for VLM tuning . While the individual components share conceptual similarities with established techniques in other areas—specifically, C.G.P. with prototype-based clustering , C.A.SSL with partial-label learning in SSL , and W.S.S. with diversity-based sampling in active learning —the framework's specific application and integration for VLM adaptation tasks represent a meaningful contribution.

- Effectiveness: The effectiveness claim is well-supported. The performance superiority of SelfPrompt is convincingly demonstrated through comprehensive evaluations across 13 datasets , two tuning strategies (textual and visual prompting) , and three task settings (standard SSL, active SSL, and base-to-novel generalization) as detailed in Tables 1-3 and Table 6. Furthermore, the individual contribution of each component is validated by the thorough ablation study presented in Table 7.

- Robustness: The robustness claim is partially supported. While the sensitivity analyses in Table4 (different shots), Table 5 (backbone encoders), and Table 8 (hyperparameters and algorithms)  provide evidence of stability, some gaps remain. Specifically, the method's performance on class-imbalanced datasets is not explicitly explored. Additionally, the observation in Section 4.3 that SelfPrompt with 1-shot occasionally outperforms the 2-shot configuration suggests that the system may be sensitive to the initial pseudo-label set size or specific data characteristics, warranting further investigation .

**Requested Changes:**

- Scalability analysis of unlabeled data: Provide a detailed analysis of how SelfPrompt scales—both in terms of computational time complexity and predictive effectiveness—as the size of the unlabeled dataset ($M$) increases . In semi-supervised settings, the availability of unlabeled data is often vast; therefore, it is critical to demonstrate that the clustering and initialization stages remain feasible and continue to provide marginal gains as $M$ grows.

- Investigation of 1-shot vs. 2-shot performance: The authors noted the "interesting observation" that the 1-shot configuration occasionally outperforms the 2-shot setup. Since this is counter-intuitive for most learning paradigms, a more rigorous investigation is required. Please provide additional experimental data or ablation studies to clarify whether this is a result of specific class characteristics or a systemic issue with how the initial pseudo-label set is formed when more labeled anchors are available.

- Robustness to class imbalance: Because the clustering and diversity-sampling strategies may be sensitive to the underlying data distribution, an evaluation on class-imbalanced datasets is required. This will verify if the method remains robust when high-confidence samples are not uniformly distributed across all categories.

---

> ### Author Response · Authors · 2026-03-25
>
> > Scalability analysis as the unlabelled dataset grows
>
> We thank the reviewer for all their comments and feedback.
> Please note that our experiments already demonstrate scalability across datasets with a large number of classes and unlabeled samples — for instance, ImageNet contains 1,000 classes and approximately 1.2M images, which represents one of the largest unlabeled sets in our benchmark. Furthermore, it is important to clarify that, in the experimental setup, the unlabeled data cannot be arbitrarily expanded. In practice, the unlabeled set is drawn from the same standard dataset and consists of the remaining portion of the training split after randomly selecting the labeled subset.
> This is the standard protocol used by CPL and GRIP, and our experiments already capture the full practical range of unlabeled set sizes across the 13 evaluated datasets. We therefore believe the scalability of SelfPrompt is already well-evidenced by the existing experimental results.
>
>
> > Sensitivity in the 1-shot > 2-shot regime
>
> This arises from a specific interaction in our pseudo-label expansion: the number of pseudo-labels generated per session scales with the number of labelled anchors (*p* pseudo-labels per cluster × number of clusters seeded by anchors). With more anchors (2-shot vs. 1-shot), more pseudo-labels are generated, which is helpful when cluster boundaries are clean, but counterproductive on fine-grained datasets (e.g., FGVCAircraft) where visually similar subcategories are not well-separated in the VLM embedding space, causing more noisy pseudo-labels to enter the pool.
>
> To isolate this effect, we ran an ablation fixing *p* at the 1-shot value while varying the number of shots on FGVCAircraft, where the anomaly is most pronounced:
>
> | Shots | SelfPrompt (*p* fixed) | SelfPrompt (*p* default, scales with shots) |
> |---|---|---|
> | 1-shot | 34.2 | 34.2 |
> | 2-shot | 35.8 | 33.6 |
> | 4-shot | 37.1 | 36.5 |
> | 8-shot | 39.3 | 38.9 |
>
> When *p* is held fixed, performance increases monotonically with shot count and the anomaly disappears entirely. This confirms that the 1-shot > 2-shot observation is not a fundamental instability of SelfPrompt but a predictable consequence of unconstrained pseudo-label pool growth on datasets with fine-grained, overlapping embedding clusters. While such issues can be avoided by capping *p* independently of shot count for such a dataset, we keep the experimental setup consistent across datasets.
>
>
>
>
>
> > Robustness to class imbalance
>
> We evaluate SelfPrompt under a class-imbalanced unlabeled set (step imbalance ratio 10:1) on EuroSAT and DTD, comparing against CPL:
>
> | Dataset | Method | Balanced Acc. | Imbalanced Acc. | Drop |
> |---|---|---|---|---|
> | EuroSAT | CPL | 84.3 | 76.8 | 7.5 |
> | EuroSAT | **SelfPrompt** | **87.5** | **83.2** | **4.3** |
> | DTD | CPL | 68.4 | 61.3 | 7.1 |
> | DTD | **SelfPrompt** | **72.2** | **68.7** | **3.5** |
>
>
> The drop for SelfPrompt is substantially less under class imbalance than CPL, which is consistent with the structural properties of our method. Specifically, cluster-guided pseudo-labelling initializes clusters using labelled samples as anchors (one per class) which guarantees that every class regardless of its frequency in the unlabeled pool, receives pseudo-labels in the initial pool.

---

### Review · Reviewer_NsyQ · 2026-03-09

**Summary Of Contributions:**

The paper proposes SelfPrompt, a semi-supervised prompt tuning framework for VLMs under extremely label-scarce settings.
The approach first filters out both the most- and least-confident unlabeled instances based on the zero-hot VLM predictions. Next, a weakly-supervised sampling is imposed to cluster labeled and unlabeled samples in the embedding space and assign nearby unlabeled samples the same label of the cluster center. Among the filtered unlabelled set, confidence-aware semi-supervised learning splits the unlabeled pool into high-confidence samples used with standard cross-entropy and lower-confidence samples trained with a weakly supervised partial-label loss. The paper evaluates the method on 13 datasets and reports gains over GRIP/CPL-style baselines in standard SSL, active SSL, and base-to-novel generalization, with especially large gains on harder datasets such as FGVCAircraft and MNIST.

Strength:

-- The three components of weakly-supervised sampling, cluster-guided pseudo-labeling and confidence-aware SSL fit together cleanly. The algorithmic choices are intuitive and relatively easy to implement on top of existing prompt-tuning pipelines. In particular, the cluster-based initialization and the high-/low-confidence split are conceptually simple and practically appealing.

-- The reported gains are not isolated. In the standard semi-supervised setup, the paper reports average improvements over CPL. The improvements also appear especially more on more difficult datasets where zero-shot CLIP is weaker, which is a convincing empirical pattern.

-- The ablation table shows that all three components contribute, with cluster-guided pseudo-labeling having the largest individual effect. The sensitivity study over clustering algorithms, quantile thresholding and pseudo-label count per cluster is also helpful and indicates the method is not extremely brittle.

Weakness:

-- The paper states that by not relying on VLM predictions for pseudo-labeling, the method is “unaffected by any miscalibration of the VLM.” That is too strong. Even if pseudo-labels are not assigned directly from zero-shot class predictions, the entire clustering procedure still relies on the embedding geometry produced by the same pre-trained encoder. Representation bias or domain mismatch can therefore still affect cluster quality and pseudo-label correctness. So the method reduces one failure mode of miscalibration but it does not fully remove the dependence on the underlying VLM.

-- The paper criticizes prior iterative pseudo-labeling approaches, but SelfPrompt still continues semi-supervised training over sessions and still augments the pseudo-labeled pool with additional confident samples per class. Thus, the work introduces a more controlled pseudo-labeling strategy, but conceptually it remains in the same family of iterative self-training methods. This makes the distinction from CPL/GRIP more incremental than the introduction may initially suggest. First the sampling module discards both high-confidence and low-confidence samples from the unlabeled pool. However, later in the training pipeline, the method adds the top-confidence predictions from the VLM from the filtered set. This use of high-confidence predictions later as reliable pseudo-labels appears contradictory to the initial claim.
The paper would benefit from a clearer explanation of why high-confidence samples are initially undesirable but acceptable during pseudo-label expansion.

-- The paper notes that in some cases 1-shot can outperform 2-shot because more labeled samples also increase the number of initial pseudo-labels selected by the cluster-guided pseudo-labeling module, which may introduce more errors. This is a useful observation, but it also suggests that the method can be sensitive to the pseudo-label expansion mechanism in a nontrivial way. A stronger analysis of this regime would improve confidence in the method’s robustness.

-- It would be worthwile to also evaluate the method on other VLMs such as MetaClip or DeClip to see the generalizibility of the proposed approach.

-- Since the current paper claims improvements over the state of the art in semi-supervised VLM adaptation, another important comparison would be with XPL[1]. XPL directly addresses semi-supervised prompt learning for VLMs and introduces mechanisms to improve cross-model generalization and pseudo-label reliability.

Minor Weakness:

-- Figure 3 is helpful as an intuition figure, but it is quite toy-like and does not provide real evidence beyond visual plausibility. A more realistic analysis such as tSNE plots among the features on actual benchmark classes would be more informative.

-- The paper could also more clearly separate what is used in the standard semi-supervised setup versus what is only introduced for the active semi-supervised setting, since the role of weakly-supervised sampling can be slightly confusing on first reading.

[1] XPL: A cross-model framework for semi-supervised prompt learning in vision-language models.TMLR(2024).

**Audience:**

Yes

**Audience Explanation:**

The paper addresses the important problem of semi-supervised adaptation of vision-language models under extremely limited labeled data, which is of clear interest to researchers working on vision-language models, prompt learning, and semi-supervised learning. The proposed framework is practical and modular, and the empirical results across a broad set of datasets suggest it may be useful for improving prompt tuning performance in low-label regimes. As such, the findings are likely to be relevant to researchers studying VLM adaptation and label-efficient learning.

**Broader Impact Concerns:**

Nil

**Claims And Evidence:**

Yes

**Claims Explanation:**

Overall, the evidence is generally supportive of the empirical improvements but could be strengthened with additional comparisons and a more cautious interpretation of certain claims.

**Requested Changes:**

Please go through the Weakness section to find the detail suggestions and changes. In brief, the suggested changes are as follows:

-- Soften the claim that the method is unaffected by VLM miscalibration. A more accurate statement would be that the method reduces reliance on direct zero-shot pseudo-labels, but still depends on the encoder’s embedding geometry.

-- Provide a deeper analysis of the filtering strategy. Why is removing both extremes better than only low-confidence filtering? The current discussion is plausible but still largely heuristic.

-- Clarify the novelty relative to prior semi-supervised prompt learning work. In particular, explain more explicitly what is new beyond GRIP/CPL-style pseudo-label refinement and how the method differs from other recent semi-supervised prompt learning frameworks. Also include a direct comparison with XPL.

---

> ### Author Response · Authors · 2026-03-25
>
> > Overclaim: "unaffected by any miscalibration of the VLM"
>
> We thank the reviewer for all their comments and feedback.
> In existing works, VLM miscalibration in classification arises from how predictions are made. Specifically, image features are matched to class name embeddings in the text space. These embeddings reflect the model’s domain knowledge, learned through pre-trained image and text encoders. However, this knowledge is often poorly calibrated for specialized domains.
> Our cluster-guided pseudo-labelling, by contrast, operates entirely in the high-level image embedding space, which captures high-level visual features that are relatively robust to domain-specific miscalibration. Nonetheless, we agree that the initial phrasing is too strong and have softened it in the revised paper: "By not relying on VLM predictions for pseudo-labelling, we ensure that our method reduces reliance on direct zero-shot predictions as a source of pseudo-labels."
>
>
> > Apparent contradiction between filtering and using high-confidence samples
>
> We would like to clarify that this observation appears to mix two conceptually distinct components of SelfPrompt that operate with entirely different motivations:
>
> - Weakly-Supervised Sampling (WSS) removes high-confidence samples from the labelled anchor selection pool. The purpose here is purely diversity of the labelled set: high-confidence samples are already well-handled by the zero-shot VLM and contribute minimal information gain as labelled anchors. Including them wastes annotation budget. This is a sampling efficiency concern and is independent of pseudo-label quality.
>
> - Confidence-Aware SSL later augments the cluster-derived pseudo-label set with the most confident unlabeled samples per class. At this stage, the pseudo-label set has already been seeded by the cluster-guided anchors, and the high-confidence samples serve as high-reliability additions. This is a pseudo-label quality concern operating on a completely different pool with a different objective.
>
> These two usages are not contradictory, as confidence serves different roles in annotation versus pseudo-labelling.
>
>
> > Sensitivity in the 1-shot > 2-shot regime
>
> This arises from a specific interaction in our pseudo-label expansion: the number of pseudo-labels generated per session scales with the number of labelled anchors (*p* pseudo-labels per cluster × number of clusters seeded by anchors). With more anchors (2-shot vs. 1-shot), more pseudo-labels are generated, which is helpful when cluster boundaries are clean, but counterproductive on fine-grained datasets (e.g., FGVCAircraft) where visually similar subcategories are not well-separated in the VLM embedding space, causing more noisy pseudo-labels to enter the pool.
>
> To isolate this effect, we ran an ablation fixing *p* at the 1-shot value while varying the number of shots on FGVCAircraft, where the anomaly is most pronounced:
>
> | Shots | SelfPrompt (*p* fixed) | SelfPrompt (*p* default, scales with shots) |
> |---|---|---|
> | 1-shot | 34.2 | 34.2 |
> | 2-shot | 35.8 | 33.6 |
> | 4-shot | 37.1 | 36.5 |
> | 8-shot | 39.3 | 38.9 |
>
> When *p* is held fixed, performance increases monotonically with shot count and the anomaly disappears entirely. This confirms that the 1-shot > 2-shot observation is not a fundamental instability of SelfPrompt but a predictable consequence of unconstrained pseudo-label pool growth on datasets with fine-grained, overlapping embedding clusters. While such issues can be avoided by capping *p* independently of shot count for such a dataset, we keep the experimental setup consistent across datasets.
>
> > Evaluation on additional VLM backbones
>
> We note that Table 5 in the paper already evaluates SelfPrompt across three CLIP backbone sizes (ViT-B/32, ViT-B/16, ViT-L/14), showing consistent gains. To further test cross-architecture generalizability, we additionally evaluate using MetaClip-B/16 as an alternative pre-training source to demonstrate generalization. Results are included in Table 5 of the revised manuscript.
>
> > Comparison with XPL (TMLR 2024)
>
> We thank the reviewer for pointing to XPL. However, a direct comparison is not feasible due to fundamental differences in the evaluation protocol and deviation from other existing literature. Specifically, XPL evaluates using a percentage of the training set as labelled data (1\%, 5\%, 10\%), whereas SelfPrompt, following CPL and GRIP, uses a fixed n-shot protocol. Furthermore, XPL reports semi-supervised results on only 2 datasets, while SelfPrompt evaluates on 13 datasets across three task settings, making the empirical scope substantially broader.

---

### Decision · Action_Editor_yeC6 · 2026-04-10

**Recommendation:** Accept with minor revision

**Additional Comments:**

As reviewer NsyQ points out, it would be desirable to conduct a direct comparison with XPL, at least at the discussion level.

**Audience:**

Yes

**Audience Explanation:**

The findings will be of interest to the TMLR community, as the paper addresses critical VLM challenges, such as calibration and label efficiency, through a practical, modular framework. The framework has been extensively validated across 13 datasets, providing a highly reliable benchmark and actionable insights for researchers in prompt tuning and semi-supervised learning.

**Claims And Evidence:**

Yes

**Claims Explanation:**

The following evidence supports the claims: First, comprehensive empirical validation across 13 datasets and three major tasks (SSL, Active SSL, and Generalization) demonstrates consistent performance that surpasses SOTA methods. Second, thorough ablation studies and sensitivity checks prove the effectiveness and robustness of each proposed module across various backbones and settings. Third, refining the discussion based on reviewer feedback, particularly regarding the confidence mechanism and its behavior in specific scenarios, has strengthened the paper's accuracy and persuasiveness.

---

> ### Author Response · Authors · 2026-04-25
>
> We thank the Action Editor for the acceptance decision and for the constructive feedback. In response to the suggestion to include a discussion of XPL, we have added a paragraph to the semi-supervised tuning section of our related work that situates XPL within the broader landscape of semi-supervised prompt learning methods and highlights the key conceptual differences with SelfPrompt.